# Comparative Study of the Convertibility of Agricultural Residues and Other Cellulose-Containing Materials in Hydrolysis with *Penicillium verruculosum* Cellulase Complex

**Dmitrii O. Osipov** [1,*]**, Gleb S. Dotsenko** [1]**, Olga A. Sinitsyna** [2]**, Elena G. Kondratieva** [2]**,
Ivan N. Zorov** [1,2]**, Igor A. Shashkov** [1]**, Aidar D. Satrutdinov** [1] **and Arkady P. Sinitsyn** [1,2]

[1] Federal Research Centre "Fundamentals of Biotechnology", Russian Academy of Sciences,
119071 Moscow, Russia; gsdotsenko@gmail.com (G.S.D.); inzorov@mail.ru (I.N.Z.);
igorshashkov@bk.ru (I.A.S.); sat-aidar@mail.ru (A.D.S.); apsinitsyn@gmail.com (A.P.S.)

[2] Department of Chemistry, M. V. Lomonosov Moscow State University, 119991 Moscow, Russia;
oasinitsyna@gmail.com (O.A.S.); elgenkon@inbox.ru (E.G.K.)

* Correspondence: d.osipov@fbras.ru

**Abstract:** Non-edible cellulose-containing biomass is a promising and abundant feedstock for simple sugar production. This study presents the results of different cellulose-containing materials (CCM) hydrolysis experiments with *P. verruculosum* enzyme complexes in laboratory conditions. Among the non-pretreated substrates, only a few had a relatively high convertibility—soy bean husks (31%) and sugar beat pulp (20%)—while wheat straw, oat husks, sunflower peals, and corn stalks had a low convertibility of 3% to 12%. This indicates that a major part of CCM needs pretreatment. Steam-exploded (with Ca(OH)$_2$) soy bean and oat husks (76% and 58%), fine ball-milled aspen wood and nitric acid-pretreated aspen wood (62% and 78%), and steam-exploded (with sulfuric acid) corn stalks (55%) had a high convertibility. Woody biomass pretreated with pulp and paper mills also had a high convertibility (56–78%)—e.g., never dried kraft hardwood and softwood pulp (both bleached and unbleached). These results demonstrate that effective cellulose-containing material processing into simple sugars is possible. Simple sugars derived from CCM using *P. verruculosum* preparation are a promising feedstock for the microbiological production of biofuels (bioethanol and biobutanol), aminoacids, and organic acids (e.g., lactic acid for polylactic acid production).

**Keywords:** cellulose-containing materials; *Penicillium verruculosum*; recombinant enzymes; pretreatment

## 1. Introduction

All natural cellulose fibers have their origin in the plant cell wall. Different plants' cell walls are radically different, but the cellulose content remains constant in the range of 35% to 50% of plant dry weight [1]. With rare exceptions, cellulose fibers are encased in a matrix composed of hemicelluloses and lignin, which constitute approximately 20–35% and 5–30% of plant dry weight [1–3]. Plant cell wall composition and architecture are fundamental characteristics of cellulose-containing biomass. While some other characteristics, such as size and moisture content, could be easily adjusted, carbohydrate and lignin content, pore density, and cellulose crystallinity are much harder to change and make the utilization of cellulose-containing materials more complex than that of pure cellulose.

The main requirements for biotechnology feedstock are low cost, availability, and high sugar yield [4,5]. The cost and availability depend on the logistics. In turn, the sugar yield in enzymatic hydrolysis or the convertibility of cellulose-containing materials are determined by fundamental

characteristics. It is necessary to identify cellulose-containing materials that match the requirements for biotechnology processes. It is reasonable to obtain simple sugars from low-cost non-food feedstock firstly due to the lack of competition with edible biomass and secondly to resolve some ecological problems—i.e., environmental pollution, greenhouse gas emission, or the alienation of productive land for landfills.

The first group of widely available potential feedstock is agricultural waste which consists mainly of herbaceous biomass (straw, husks, cobs, peels, leaves) with a low lignin content. The worldwide production of crop plant residues is about 3.4 billion tons yearly [6], including 100 million dry tons of corn stover [7], 280 million dry tons of bagasse [8], and 354 million dry tons of wheat straw [9]. As sugarcane bagasse in tropical countries, sugar beet pulp is a major by-product of the sugar industry in countries with a continental climate. The annual production of sugar beet pulp is 21–22 million wet tons, and its main use is as animal feed [10]. More than 100 million tons of agricultural plant residues are generated every year in the Russian Federation [11]. In general, the dry weight residue/grain ratio for cereal varies from 1 to 2 [12]. This agricultural biomass requires reutilization, otherwise it is left on field, dumped, and incinerated.

The annual harvest of wood biomass for saw logs and paper and pulp is 960 million tons. The estimated forestry processing wastes in the EU are about 88.2 million tons per year [13]; in the Russian Federation, they are about 35.5 million m$^3$/yr [14]. Woodworking industry residues are widespread and often have no need to be additionally milled, but felling residues have to. Operating felling residues are important for sustainable forest use and preventing forest soil contamination. Aspen (*Populus tremula*) has a great potential in future bioeconomy as an energy crop due to its fast growth or as a felling residue because of its poor wood.

Food production generates a huge amount of residues and by-products that are usually used as animal feed due to residual protein, starch, and sugar [15,16] but could find application as a biotechnology feedstock. A total of 90 to 150 million tons of cultivated wheat is converted into bran every year [17]. Starch-containing bran was studied for bioethanol production [18] and protein extraction [19]. The main coproducts of grain-derived fuel and food-grade ethanol production are distillers' dry grain and solubles, which are limited in monogastric livestock. The growing production is expected to drive its supple up and prices down, making it a promising cellulose-containing resource [20].

It is common opinion that raw wet and dry cellulose-containing materials need to be pretreated prior enzymatic saccharification due to their rigidity. The most studied types of pretreatment are dry or wet mechanical milling, steam explosion and its varieties, dilute acid or alkaline pretreatment, and pretreatment with organic solvents. Irrespective of the pretreatment process, the cellulose-containing material's fundamental characteristics change. While relatively novel pretreatment processes require additional optimization and upscaling, pulp and paper mills are already fine-tuned for the production of delignified cellulose-rich materials located close to forest resources and could be quickly integrated into forest biorefinery. The estimated global production of mechanical and semi-chemical pulp is 36.3, with chemical pulp at 142.4, Kraft pulp (bleached and unbleached) at 140, and sulfite pulp at 2.4 million t/yr in 2017 [21]. The only technology missing in pulp mill production chain is enzymatic hydrolysis. The production of cellulosic sugars and further biofuels and biochemicals on a reorganized mill is economically feasible because of the compatible supply, facilities, and workforce [22]

In some cases, it is reasonable to process unconventional local cellulose-containing materials, such as municipal solid waste or algal biomass [23].

The aim of this research is to evaluate the enzymatic convertibility of cellulose-containing materials of different origin—including agricultural and industrial byproducts and waste in enzymatic hydrolysis with *P. verruculosum* cellulolytic complexes—to simple sugars which could be used as raw materials in different sectors of bioindustry.

## 2. Materials and Methods

### 2.1. Samples of CCM

The sugar beet pulp was from Kanevsky sugar refinery (Krasnodar, Russia) and wheat straw from Khimavtomatica SPA (Moscow, Russia). Dried and steam explosion-pretreated bagasse was provided by DNL (Rotterdam, The Netherlands), Florida Crystals (West Palm Beach, FL, USA), and BC International. Sunflower peels were from East Siberian Biotechnology Center PLC (Tulun, Russia). Raw and pretreated soy husks and oat bran were provided by ADM (Decatur, IL, USA). Dried and steam explosion-pretreated corn stalks were obtained from NREL (Golden, CO, USA), Dyadic International (Jupiter, FL, USA), and ADM (Decatur, IL, USA). Corn bran was obtained from Gruma S.A.B. de S.V. (San Pedro, Mexico). Wheat bran and brewing waste were from Bigor LLC (Moscow, Russia). Wet and dried distillers grains were provided by ADM (Decatur, Illinois). Original and ball-milled wood samples (pine, aspen, larch, hevea) were from the East Siberian Biotechnology Center (Tulun, Russia). Samples of bleached and unbleached softwood and hardwood pulp were produced from spruce and birch-aspen mixtures, respectively, from Arkhangelsk Pulp and Paper Mill (Arkhangelsk, Russia).

### 2.2. Substrates

Carboxymethylcellulose (CMC, sodium salt, medium viscosity), birch glucuronoxylan, *p*-nitrophenyl-β-D-glucopyranoside, and cellobiose were purchased from Sigma (St. Louis, MO, USA); Avicel PH105 cellulose was from Serva (Heidelberg, Germany); filter paper No.1 was from Whatman (Little Chalfont, UK).

### 2.3. Enzyme Preparations

B151 and F10 preparations represent freeze-dried ultra-concentrated cultural filtrates of *Penicillium verruculosum* strains B151 (represents cellulase and xylanase complex) [24] and F10 (recombinant strain after the heterologous expression of *Aspergillus niger* β-glucosidase (cellobiase)) cultivated in 1l fermenters at the Institute of Biochemistry of RAS (Moscow, Russia) [25].

### 2.4. Analytical Procedures

#### 2.4.1. Enzyme Activity Assays

The filter paper activity was determined using a standard method with 1 cm × 5 cm (50 mg) Whatman No. 1 filter paper strips in 0.05 sodium citrate buffer pH 4.8 at 50 °C for 1 h and DNS reagent for measuring reducing sugars [26]. The CMCase and xylanase activities were determined by a reducing sugar release at pH 5.0 and 50 °C after 10 min using a substrate concentration of 5 mg/mL in the reaction mixture [27]. Avicelase activity was determined by the reducing sugar release at pH 5.0 and 40 °C after 60 min of enzyme reaction with Avicel PH105 (5 mg/mL) [28]. Reducing sugars were assayed by the Nelson-Somogyi spectrophotometric method based on molybdenum blue formation in the reaction of molybdic acid reduction by cuprous oxide. A total of 0.2 mL of reducing sugars containing sample solution with 0.2 mL of Somogyi reagent were mixed and incubated at 100 °C for 1 h, then sequentially 0.2 mL of Nelson, 0.4 mL of acetone and 1 mL of distilled water were added. Adsorption was measured at 610 nm [29]. One unit of activity corresponded to the quantity of enzyme releasing 1 μmol of reducing sugars (in glucose equivalents) for one minute. The activity against p-NP-β-glucospiranoside was determined at pH 5.0 and 40 °C by measuring the p-nitrophenol released, as described elsewhere [28]. One β-glucosidase unit of activity is the quantity of enzyme which liberates 1 micromole of *p*-nitrophenol in one minute. Cellobiase was determined by the incubation of 2.5 mM of cellobiose solution with the enzymes at 40 °C and pH 5.0 and measuring the released glucose with the glucose oxidase/peroxidase method [30]. One unit of cellobiase activity is the quantity of enzyme which liberates 1 micromole of glucose in one minute. The activities of the enzyme preparations are

given in the Table 1. Protein concentration was determined according to Lowry protein assay [31] using bovine serum albumin as a standard.

**Table 1.** Properties of *P. verruculosum* enzyme preparations (EP).

| EP | Protein, mg/g | FP, U/g | Avicel, U/g | CMC, U/g | pNPG, U/g | Cellobiose, U/g | Xylan, U/g |
|---|---|---|---|---|---|---|---|
| B151 | 970 ± 26 | 760 ± 25 | 578 ± 17 | 16,542 ± 340 | 1074 ± 52 | 603 ± 47 | 17,532 ± 505 |
| F10 | 655 ± 14 | 147 ± 5 | 853 ± 21 | 7007 ± 73 | 39,852 ± 926 | 46,663 ± 1134 | 3800 ± 39 |

### 2.4.2. Enzymatic Hydrolysis

CCM hydrolysis tests were conducted in 50 mL vessels incubated at 50 °C and 250 rpm on an Innova 40 shaker (Edison, NJ, USA) for 48 h. A reaction mixture of 20 mL total contained 100 g/L of CCM dry matter in 0.1 M of Na-acetate buffer at pH 5.0 with 10 mg/g of dry matter protein loading of B151 enzyme preparation. To overcome the cellobiose inhibition effect, an amount (40 U/g dry substrate) of cellobiase activity of F10 enzyme preparation was added to the mixture. Then, 1 mM of NaN$_3$ and 100 µg/mL of ampicillin were used to prevent contamination. Nelson-Somogyi assay [29] was used to determine the concentration of reducing sugars in the reaction mixture.

CCM enzymatic convertibility was defined as a degree of conversion (48 h) to reducing sugars (in glucose equivalent) as a percentage per initial concentration of substrate (*w/w*).

### 2.5. Pretreatment Conditions

Wheat straw; sunflower peels; bagasse; and aspen, larch, and pine sawdust were fine milled using an AGO-2 laboratory ball mill (<20 µm). Sugar beet pulp were pretreated in a double-screw extruder at 182 °C, 30 atm, for 50 s.

Aspen wood was milled (300 µm sieve) using an IM450 industrial impeller mill (SE TechPribor, Shchyokino, Russia). Milled aspen wood was pretreated in different ways: with diluted (0.9–12.7%) sulfuric acid at 120–180 °C for 15 to 180 min, with 0.18–0.54% sulfuric acid solution in ethanol and butanol, and with diluted (0.15–4.8%) nitric acid at 100–160 °C and elevated pressure for 60 min. The diluted acid pretreatment was conducted in a 100 mL pressurized steel cylinder. A cylinder with milled aspen wood and an acid solution with a solid-to-liquid ratio of 1:5 was placed in a temperature-controlled oil bath. Elevated pressure was created by nitrogen injection. By the end of the pretreatment process, the reactors were cooled in cold water, then the slurry was filtered and the remaining solids were washed with water or alcohol–water mixture.

## 3. Results and Discussion

Unpretreated agricultural residues are resistant to cellulolytic enzymes and characterized by a low reducing sugar yield in hydrolysis with a mixture of *P. verruculosum* B151 cellulase complex and F10 β-glucosidase. The wheat straw convertibility was 12%. For the sugar beet pulp, it was 20%, for oat husks it was 5%, for sunflower peels it was 3%, for corn stalks it was 10%, and for bagasse it was 17%. Only the soy husk convertibility was relatively high, at 31% (Table 2).

**Table 2.** Convertibility of different cellulose-containing materials in hydrolysis by a mixture of *P. verruculosum* B151 cellulase complex and F10 β-glucosidase.

| Substrate | Convertibility, % |
|---|---|
| **Agricultural residues** | |
| Wheat straw | 12 |
| Wheat straw pretreated by dry fine ball-milling (<20 µm) | 45 |
| Wheat straw pretreated by 1% NaOH, 85 °C | 55 |
| Wheat straw pretreated by steam explosion | 75 |
| Wheat straw steam pretreated by steam explosion with Ca(OH)$_2$ | 69 |

**Table 2.** *Cont.*

| Substrate | Convertibility, % |
|---|---|
| Sugar beet pulp | 20 |
| Sugar beet pulp extruded | 27 |
| Oat husks | 5 |
| Oat husks pretreated by steam explosion with $Ca(OH)_2$ | 76 |
| Soy bean husks | 38 |
| Soy bean husks pretreated by steam explosion with $Ca(OH)_2$ | 58 |
| Sunflower peels | 3 |
| Sunflower peels pretreated by dry fine ball-milling (<20 μm) | 7 |
| Corn stalks | 10 |
| Corn stalks pretreated by steam explosion with $H_2SO_4$ | 55 |
| Corn stalks pretreated by steam explosion with $Ca(OH)_2$ | 36 |
| Sugar cane bagasse | 18 |
| Sugar cane bagasse pretreated by dry fine ball-milling (<20 μm) | 42 |
| Sugar cane bagasse pretreated by steam explosion with $Ca(OH)_2$ | 41 |
| Sugar cane bagasse pretreated by steam explosion with $H_2SO_4$ | 34 |
| **Food-industry waste** | |
| Brewing waste (rye-wheat) | 10 |
| Wheat bran (destarched) | 14 |
| Corn bran (destarched) | 12 |
| Distillers grains wet (WDG) | 18 |
| Distillers grains dried (DDG) | 16 |
| **Pulp and paper industry products** | |
| Never dried bleached softwood kraft pulp | 78 |
| Never dried unbleached softwood kraft pulp | 68 |
| Dried bleached softwood kraft pulp | 58 |
| Dried unbleached softwood kraft pulp | 48 |
| Never dried bleached hardwood kraft pulp | 66 |
| Never dried unbleached hardwood kraft pulp | 56 |
| Dried bleached hardwood kraft pulp | 50 |
| Dried unbleached hardwood kraft pulp | 42 |
| **Wood industry waste and forestry residues** | |
| Pine sawdust | 8 |
| Pine sawdust (deresinated) pretreated by dry fine ball-milling (<20 μm) | 45 |
| Larch sawdust | 6 |
| Larch sawdust pretreated by dry fine ball-milling (<20 μm) | 22 |
| Aspen sawdust | 8 |
| Aspen sawdust pretreated by dry fine ball-milling (<20 μm) | 50 |
| Hevea sawdust | 4 |
| Hevea sawdust pretreated by dry fine ball-milling (<20 μm) | 14 |

Mechanical pretreatments such as fine ball milling or extruding have resulted in increasing the convertibility of these substrates 1.35–3.75 fold. The data of enzymatic hydrolysis obtained for mechanically pretreated materials have shown that the ball milling (which gives an average particle size of less than 20 μm) of wheat straw and bagasse results in increased convertibility to 45% and 42%, respectively. The ball milling of sunflower peels leads to a very limited improvement in convertibility at 7%, which indicates that they are practically not digestible by cellulolytic enzymes. Sugar beet pulp pretreatment by extrusion has also shown a limited improvement in convertibility (to 27% only).

The wheat straw convertibility after delignification with hot alkaline solution increased 4.6-fold (up to 55%).

Steam explosion pretreatment with different additives ($H_2SO_4$, $Ca(OH)_2$) demonstrated that wheat straw, out husks, soy bean husks, corn stalks, and bagasse were easy to hydrolyze with enzymes, and this pretreatment enhanced the convertibility up to 69–75%, 76%, 58%, 36–55%, and 34–41%, respectively. Supplementation with $H_2SO_4$ and $Ca(OH)_2$ has shown an opposite result for different materials: calcium hydroxide is preferable for bagasse pretreatment (7% higher sugar yield), while a corn stalk pretreatment required sulfuric acid (19% higher sugar yield). Wheat straw steam pretreatment required no additives.

The enzymatic hydrolysis of unpretreated food-industry waste has shown that this kind of cellulose-containing material is far from being a potential source of simple sugars for biotechnology. Thus, the convertibility of brewing waste, destarched corn, and wheat bran was very low, at 10%, 12%, and 14%, respectively. The convertibility of wet and dry distillers grains hydrolysis was slightly higher, at 18% wet and 16% dry, respectively.

Pulp and paper production is a large-tonnage and streamlined industry. The range of products in this area is very wide. They differ in the raw materials (hardwood or softwood, others) used and the way they are produced (wood cooking, bleaching). Creating an integrated biorefinery plant around existing pulp and paper mills would enhance their marketability by efficient converting all biomass components into value-added products [32]. The Kraft pulping process is a promising pretreatment technology for biocatalytic conversion of cellulose and hemicelluloses to glucose and other monosaccharides [33,34]. The convertibility of newer dry as well as dried kraft fibers representing by bleached and unbleached soft wood and hardwood pulp was evaluated (Table 1). The highest convertibility was demonstrated by wet bleached softwood pulp, at 78%; the reason for the high convertibility is the almost complete removal of lignin by the Kraft process (a remaining lignin content was 2–3% [33]). The convertibility of wet bleached hardwood pulp was 58%; this is lower compared with bleached softwood pulp because of the xylan influence [33]. Unbleached wet softwood and hardwood pulp had approximately a 1.1 times lower convertibility compared with similar types of bleached pulp; the decrease in convertibility of unbleached pulp is explained by the higher lignin content.

The drying and subsequent hornification of all kraft pulps types had a significant effect on convertibility as it reduces swelling and the cellulose fiber accessibility [35] and causes a collapse in the pore structure [36]. The convertibility was reduced by 1.3–1.4 times compared to wet pulp.

The recycling and utilization of wood industry wastes and forestry residues is crucial for wood processing and environmental security. In total, five types of wood species were included in this study. The convertibility of pine, larch, aspen, and hevea sawdust was low and found to be in the range of 4–8% (Table 1). Mechanical pretreatment (dry fine ball-milling that results in an average particle size less than 20 μm) has resulted in a significant increase in the convertibility of pine wood and aspen wood to up to 45–50%; in the case of larch, it is up to 22%. The increase in convertibility was due to defibrillation and reduction in the crystallinity of fibers and increasing surface area related to reducing particle size [35]. Despite its high efficiency, fine ball-milling has serious disadvantages, such as being energy consuming and difficulties in scaling up [37]. In view of the rising energy prices and power intensity, fine milling is not economically reasonable [38].

To counteract these disadvantages, less intensive milling processes can be combined with chemical and physicochemical pretreatments such as dilute acid and organosolv pretreatments. We have studied the convertibility of aspen wood subjected to pretreatment by different water and organic solutions of mineral acids. Relatively low temperatures of pretreatment process were selected to prevent the unfavorable degradation of carbohydrates and inhibitors formation.

Pretreatment of aspen wood by dilute acids: Five samples were obtained using dilute sulfuric acid pretreatment of aspen wood particles (200–300 μm). The results demonstrate a linear correlation between acid concentration and substrate convertibility (Table 3). There was no difference found (58.4% convertibility) for samples processed with 12.7% and 8.7% sulfuric acid, which could mean that the maximum available polysaccharides for enzymatic hydrolysis due to the solubilization of hemicelluloses are limited. Further reduction in the acid concentration to 4.4% and 1.8% results in the

convertibility decreasing to 44.2% and 42.9%, respectively. Further reduction in the acid concentration to 0.9% provides a convertibility of 41.6%, which is just 1.4 times lower than the result obtained by 12.7% acid. Such a reduction in chemical consumption can be economically feasible even with a lower biomass convertibility.

**Table 3.** Convertibility of aspen wood pretreated using different types of acid-containing solutions at elevated temperatures.

| Conditions | | | | Convertibility, % |
|---|---|---|---|---|
| Aspen wood (200–300 μm) | | | | 8 |
| **Dilute sulfuric acid pretreatment** | | | | |
| Temperature, °C | Time, h | Acid concentration, % | | |
| | | 12.7 | | 58.4 |
| | | 8.7 | | 58.4 |
| 140 | 1 | 4.4 | | 44.2 |
| | | 1.8 | | 42.9 |
| | | 0.9 | | 41.6 |
| **Dilute nitric acid pretreatment** | | | | |
| Temperature, °C | Time, h | Pressure, at | Acid concentration, % | |
| 100 | | 6 | 4.8 | 61.2 |
| 130 | | 5 | 1.1 | 62.8 |
| 125 | | 9 | 4.8 | 60.6 |
| 125 | | 14 | 4.8 | 65.1 |
| 125 | | 18 | 4.8 | 78.7 |
| 125 | 1 | 22 | 4.8 | 78.6 |
| 150 | | 5 | 0.2 | 49.3 |
| 160 | | 5 | 0.5 | 45.6 |
| 160 | | 5 | 0.7 | 48.2 |
| 160 | | 5 | 0.3 | 46.9 |
| 160 | | 5 | 0.2 | 43.7 |
| **Sulfuric acid organosolv** | | | | |
| Temperature, °C | Time, h | Organic phase, % | Acid concentration, % | |
| | | 50% EtOH | 0.36 | 38.7 |
| | | 65% EtOH | 0.54 | 48.3 |
| | | 80% EtOH | 0.54 | 43.9 |
| 140 | 1 | 50% BtOH | 0.36 | 36.4 |
| | | 65% BtOH | 0.54 | 58.7 |
| | | 80% BtOH | 0.54 | 51.1 |
| | | 25% EtOH, 25% BtOH | 0.54 | 54.3 |
| | | 40% EtOH, 10% BtOH | 0.54 | 51.1 |
| | | 20% EtOH, 40% BtOH | 0.36 | 51.5 |
| | | 20% EtOH, 40% BtOH | 0.18 | 45.9 |
| | | 25% EtOH, 25% BtOH | 0.18 | 37.6 |
| | | 10% EtOH, 40% BtOH | 0.18 | 36.0 |

Enhancing the hydrolysis of lignocellulose biomass for the efficient conversion of cellulose and hemicellulose by pretreatment using nitric acid has not been highly studied compared to sulfuric acid. To estimate the influence of nitric acid on the convertibility of 200–300 μm aspen wood in severe and mild conditions, another series of experiments was carried out.

The experimental results demonstrate that the reducing sugars yields after a relatively mild pretreatment at 100–130 °C, 5–6 at, and 1–4.8% nitric acid were purely comparable, at about 60% (Table 3). Maximum convertibility was achieved with a subsequent elevation of gas pressure. The aspen wood convertibility enhanced significantly from 60.6% to 78.7% as the pressure increased from 9 to

18–22, while the temperature and acid concentration remained constant (125 °C, 4.8%). These nitric acid concentration and pressure values found in this study are optimal for pretreatment, since increasing the temperature up to 160 °C results in a convertibility below 50% for acid concentrations of 0.3–0.7%. There was no additional effect on the reducing sugars yield when the pressure was raised from 18 to 22 at.

It could be concluded that aspen wood pretreatment with dilute nitric acid (convertibility 78.7% at an acid concentration of 4.8%) is more efficient than with dilute sulfuric acid (convertibility 58.7% at an acid concentration of 12.7%) but requires more complex equipment.

Organosolve pretreatment of aspen wood: The results obtained in this study demonstrate that dilute acid pretreatment is very effective. However, after such pretreatment lignin solubilized poorly [39], even though the hemicellulose matrix is being dissolved. Thus, the next panel of experiments was aimed to discover best conditions for fractionation and recovery of lignin, cellulose, and hemicelluloses. All the experiments were conducted at constant temperature 140 °C for the same time 1 h, but at different concentrations of organic solvents (ethanol and *n*-buthanol) and sulfuric acid (0.18–0.54%) as a catalyst.

The data display that 50% (*v/v*) alcohol–water mixtures have the same efficiency of 36–37% with 0.36% acid. Using a 0.54% acid concentration, 65% *n*-butanol is more preferable than 65% ethanol, ensuring a convertibility of 58.7% and 48.3%, respectively (Table 3). Organosolve pretreatment with 80% alcohol results in a lower convertibility (43.9% and 54.1% for ethanol and *n*-butanol respectively), but *n*-butanol still provided a better enzymatic digestibility. This likely could be explained as the swelling of cellulose fibers decreased as the ethanol concentration increased [40].

During organosolve pretreatment with *n*-butanol, three fractions were obtained: black liquor containing dissolved lignin, hemicellulose-enriched liquid fraction, and cellulose-containing solid fraction [41]. This spatial separation also could explain better biomass hydrolysability after *n*-butanol pretreatment. Previously [42], it has been found that the swelling of cellulose in organic solvent strongly depends on the species of organic solvents—the solvent basicity, the molar volume, and the hydrogen bonding capability—thus, *n*-butanol is more significant in two-component mixtures. This was shown in experiments with alcohol concentrations of 25% EtOH + 25% BtOH (convertibility 54%) and 40% EtOH + 10% BtOH (convertibility 51%). However, in general, more concentrated *n*-butanol single-component mixtures are preferable to two-component mixtures.

Decreasing acid concentrations from 0.36% to 0.18% resulted in a lower convertibility using the same mixture composition (20% EtOH + 40% BtOH)—52% and 46%, respectively. At an acid concentration of 0.18%, this mixture composition has no effect on the convertibility.

These results indicate that alcohol-water mixtures allow using less concentrated acids, while biomass components could be fractionated and alcohols could be recirculated. In this study, the optimal conditions for pretreatment were found: 140 °C, 1 h, 0.54% sulfuric acid, 65% *n*-butanol. These conditions lead to a 7.4-fold increase in the convertibility or aspen wood (59% compared to 8% of untreated substrate).

## 4. Conclusions

In total, the convertibility in enzymatic hydrolysis by a mixture of *P. verruculosum* B151 cellulase complex and F10 β-glucosidase of 69 samples of original and pretreated CCM was tested in this study. A major part of the non-pretreated substrates had a low convertibility of 3% to 18%. There were only a few substrates with a higher convertibility among the original unpretreated samples—e.g., soy bean husks had a convertibility of 31%, with sugar beat pulp at 20%. The low level of convertibility of CCM indicates the necessity of pretreatment [43,44].

Among the pretreated feedstocks, steam-exploded (with Ca(OH)$_2$) soy bean and oat husks (76% and 58%), fine ball-milled aspen wood and nitric acid-pretreated aspen wood (62% and 78%), and steam-exploded (with sulfuric acid) corn stalks (55%) had a sufficiently high convertibility. It should be noted that types of cellulosic feed stocks that are the source of pulp and paper industry, such as newer

dried kraft hardwood and softwood pulp (both bleached and unbleached), had a high convertibility (56–78%). This kind of CCM from our point of view had matching characteristics with the requirements for biomass enzymatic conversion to simple sugars and downstream processes, since they have a high convertibility, bulk availability, and waste disposal problems. The softwood and hardwood kraft pulp had a higher convertibility and are the most promising types of cellulose-containing materials from our point of view.

**Author Contributions:** Conceptualization, A.P.S.; Data curation, D.O.O. and G.S.D.; Formal analysis, D.O.O., G.S.D., and I.N.Z.; Investigation, D.O.O., G.S.D., E.G.K., I.A.S., and A.D.S.; Methodology, D.O.O., O.A.S., and I.N.Z.; Supervision, A.P.S.; Writing—original draft, D.O.O.; Writing—review and editing, A.P.S. All authors have read and agreed to the published version of the manuscript.

**Funding:** The work was partially supported by the Russian Fund of Basic Research (#18-54-80027).

**Conflicts of Interest:** The authors declare no conflict of interest. The funders had no role in the design of the study; in the collection, analysis, or interpretation of data; in the writing of the manuscript; or in the decision to publish the results.

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
