# Peer review of "Comparative Study of the Convertibility of Agricultural Residues and Other Cellulose-Containing Materials in Hydrolysis with Penicillium verruculosum Cellulase Complex"

_agronomy, doi:10.3390/agronomy10111712_

Round 1

Reviewer 1 Report

In this study, the authors studied the convertibility of a few cellulose-containing raw materials using enzymatic hydrolysis. The study presents certain interest to the readers, however it needs to be further clarified.

  1. The background was well stated. However, the objective and the potential application of this study are not clear. Does this study have a potential in green alcohol production for green chemistry, such as bioplastics or others? does it offer raw materials for food and chemical industries? it needs to add these statements. 
  2. The section of Materials and Methods must be improved. 2.3 enzyme preparation is not clear. The authors should articulate the source of enzymes. Do they come from suppliers or are they made at the authors' lab? There is no description on the convertibility. Is it based on the ratio (w/w) of glucose and raw material. The 2.5 pretreatment conditions may be put in front of 2.3.
  3. The enzymatic hydrolysis, such as glucose yields, may be compared using graphs. There is no related data.
  4. Currently the compostibility of plastics or other industrial products is of intense interest. Could this study be applied to the compostibility of bioplastics? because lot of them contain cellulose biomass in their formula. If yes, it is better to add a discussion and background information in the result, discussion, conclusion and background sections.

Reviewer 2 Report

Temat artykułu jest aktualny i ważny. W artykule przedstawiono wyniki hydrolizy materiałów zawierających celulozę z kompleksami enzymatycznymi w warunkach laboratoryjnych. Chociaż prezentacja może mieć dużą wartość w praktyce ogólnej i naukowej, przed publikacją należy wziąć pod uwagę i uwzględnić w tekście szereg kwestii.

  1. W podsumowaniu pracy brakuje informacji o tym, o czym jest ta praca;
  2. Pochodzenie tych materiałów jest szczegółowo wskazane w „materiałach badawczych”, ale brak jest informacji o ich właściwościach;
  3. Rękopis nie ma określonego celu pracy;
  4. W sekcji 2.4.1 opisz krótko, czym jest „standardowa metoda z Whatman nr 1”; Należy również wskazać, jaka jest metoda „Spektrofotometryczna metoda Nelsona-Somogyi” - jest to niezbędne, aby praca była czytelna dla każdego odbiorcy;
  5. W dyskusji nad wynikami autorzy wspominają o odpadach z przemysłu spożywczego - co należy rozumieć przez to pojęcie? Dlaczego nie wspomniano o tym w opisie materiałów badawczych?
  6. Zdanie jest niezrozumiałe. Proszę wytłumacz.
  7. Wiersz 205- autorzy powinni podać bardziej szczegółowe informacje;
  8. Wnioski nie porównują materiałów i stwierdzenia, które zdaniem autorów było najlepsze;

Round 2

Reviewer 1 Report

Format editing may be needed.

Author Response

Dear reviewer,

changes have been made.

Reviewer 2 Report

The topic of the article is current. The article presents the results of materials containing cellulose, hydrolysis with enzymatic complexes in laboratory conditions. With the comments taken into account, the presentation has potential and high value in general and scientific practice. The authors referred to the suggestion in the previous review. I believe that the manuscript is acceptable in this form.

Author Response

Dear reviewer,

changes have been made.